# Hydrogeotechnical Predictive Approach for Rockfall Mountain Hazard Using Elastic Modulus and Peak Shear Stress at Soil–Rock Interface in Dry and Wet Phases at KKH Pakistan

**Ehtesham Mehmood [1],\*, Imtiaz Rashid [1], Farooq Ahmed [2], Khalid Farooq [1], Akbar Tufail [1] and Ahmed M. Ebid [3]**

1   Department of Civil Engineering, University of Engineering and Technology, Lahore 54890, Pakistan
2   Department of Geological Engineering, University of Engineering and Technology, Lahore 54890, Pakistan
3   Department of Structural Engineering, Future University in Egypt, New Cairo 11865, Egypt
\*   Correspondence: ehteshammehmood@gmail.com

**Abstract:** Predicting the susceptibility of rockfall mountain hazards for block-in-matrix soils is challenging for critical steep cuts. This research illustrates a hydrogeotechnical approach for the prediction of rockfall triggering by performing laboratory tests on low-cohesive-matrix soil collected from steep slopes with 85° to 88° angles at the Tatta Pani site, Karakorum Highway (KKH), and then real-scale moisture-induced rockfall was conducted on site for the validation of laboratory data. Laboratory data of forty quick direct shear tests on samples collected from the field depicted a 3-fold drop in peak shear stress ($P_S$) at the soil–soil interface and a 9.3-fold drop at the soil–rock interface by varying the moisture content from 1% (taken as dry phase) to a critical laboratory moisture content $(MC)_{LC}$ of 21% (taken as wet phase). Similarly, a drop in the elastic modulus ($E_S$) was observed to be 5.7-fold at the soil–soil interface and 10-fold at the soil–rock interface for a variation of moisture content from 1 % to 21% for the matrix with a permeability (k) range of $3 \times 10^{-4}$ to $5.6 \times 10^{-4}$ m/s, which depicts the criticality of moisture content for the rockfall phenomenon. The critical moisture content evaluated in laboratory is validated by an innovative field-inundation method for thirty-two moisture-induced real-scale forced rockfall cases, which showed the rock-block triggering at field dry density $(\gamma_d)_f$ and the critical field moisture content $(MC)_{FC}$ of the matrix ranging from 1.78 g/cm$^3$ to 1.92 g/cm$^3$, and 1.3% to 25.4%, respectively. Hydrogeotechnical relations, i.e., MC versus $P_S$ and $E_S$, at the soil–rock interface are developed for the prediction of rockfall triggering. The proposed correlations may be helpful in the prediction of rockfall hazards by using expected rainfall in the field for disaster warning and landslide disaster prevention at ecological geotechnical engineering projects. The results revealed that the critical $(MC)_{FC}$ and $(MC)_{LC}$ are within 20%, depicting a good confidence level of the outcomes of this research.

**Keywords:** rockfall mountain hazard; peak shear stress; elastic modulus; ecological geotechnical engineering

## 1. Introduction

The rock blocks embedded in weak soil matrix form rock blocks and soil mixtures called block-in-matrix soils [1,2]. Rockfall presents the natural event of fast down-slope freefall or the rolling movement of blocks detached from soil–rock mixtures, causing hazards to transportation corridors [3,4]. The rockfall phenomenon is attracting researchers due to its resultant sudden and frequent damage to infrastructures. Rockfall is an unstable process and severe hazard that is caused mostly by rainwater in mountainous regions, presenting an economic and social issue [5]. Soil–rock inhomogeneous mixture is frequently encountered in mountainous ranges showing weak strata prone to slope failures due to the moisture variation in strata [6,7]. The rock blocks and soil particles show different physical and strength properties in response to the loading and moisture variation in strata where the rock blocks have very high strength as compared to the soil matrix [8].

The flow of matrix material can be analyzed by topographical, hydrological, and geotechnical parameters, and it can be observed in extreme topography transition zones and regions with significant elevation differences [9]. The eco-safety of pavement infrastructures and human life in critical rockfall areas having typical extreme topography transition zones with enormous elevation differences necessitates the evaluation of the moisture strength criterion in rockfall areas.

The hydrological effect is a significant cause of the fall of boulders embedded in the matrix and the ultimate phenomenon of slope failure [10]. The flow of rock blocks and fragments, along with soil debris due to high moisture infiltration, is a complex mechanism that is still poorly understood [11]. Rockfalls are slope failures where rock and/or soil rapidly fall down, causing severe hazards to people and transport after the detachment of embedded rock blocks from the steep rock face [12]. Rockfall is the result of changes in the environmental factors along the steep cut slope portions of transport corridors, and it is considered as a significant hazard that frequently occurs on steep slopes, mainly due to rainfall [13]. Rockfall is dangerous for infrastructures and ecology along corridors and at the path of the trajectory of rock-block movement, which necessitates landslide management [10]. The initiation of rapid rockfall and rock fragments causes severe damage to structures and lifelines, where geological and geotechnical data become essential for the significant prediction of the failure of the matrix and the initiation of rock-block movement [3].

Water flow channels affect the saturation phase of matrix material, which ultimately initiates the flow of the matrix, along with the detachment of rock blocks and rock fragments embedded in the matrix [14].

The selection of predictor factors is the most significant part of the matrix failure and rockfall [15]. Rockfall is initiated by earthquakes, animals, excavations, rock slope cutting, and most frequently by the interaction of fine materials of matrix soil with high-intensity rainwater. Rainfall is the key parameter that affects the strength of the soil–rock mixture and is considered as a critical cause for disaster in landslides [16]. Rainfall is strongly related to landslides in steep-slope areas due to critical level of moisture content absorbed by soil strata, i.e., the soil matrix [17–19]. The threshold of critical moisture content is the most considerable aspect for the triggering of rockfall and rock-block movement from the saturated condition of the matrix [20,21]. The most critical factor that has been frequently observed worldwide in the occurrence of rockfall and the triggering of rock-block movement is rainfall, as it increases the moisture content and decreases the shear strength of the soil matrix in soil–rock mixtures to a critical level, causing the initiation of rock-block movement [22–24]. The pore water pressure of the soil of the slope increases due to intense rainfall, which decreases the shear strength of the soil, resulting in the detachment of rock blocks from the matrix [25–27].

The penetration of water in the matrix depends upon its permeability, which plays a key role for the softening of matrix material during and after rainfall in the area [28,29]. The permeability of soil–rock mixtures is around $3 \times 10^{-6}$ m/s with 20% small rock blocks [16], which appreciably varies with 50% small rock blocks [30] and may reach to very low levels, i.e., $1.5 \times 10^{-9}$ m/s [31]. The coefficient of permeability (k) varies with grain sizes of the soil matrix, as the permeability (k) value increases with the increase in grain size and porosity [32]; hence, the grain size and shape of the particles significantly affect the permeability of strata [16].

Soil porosity, strength, and compactness are significant factors for rockfall in block-in-matrix soil. The higher porosity favors the rapid rockfall phenomenon during rainfall. The higher level of strength and compactness of the soil strata depict the stable rock block-in-matrix soil. Hence, the evaluation of soil porosity, strength, and compactness of block-in-matrix soil are significant factors for the contact force between soil and rock and the triggering of rockfall.

The seepage of water at the soil–block interface is more critical as compared with seepage in the soil matrix during intensive rainfall [5]. The rainfall causes the softening of

geomaterials of the slope at the contact surface of the blocks and initiates the gap between block and geomaterials [27]. The adhesive forces at the soil–rock interface are weakened by the strong seepage forces of water, resulting in a decrease in the friction angle between the block surface and soil [16,27,33]; hence, the block is detached from the matrix and falls freely towards the toe of the cut slope.

Rainfall intensity can be evaluated by statistical data analysis for the considered area on the basis of rainfall history for the determination of rainfall-induced slope failures and rock-block fall [5,10,34].

Water flow patterns in the area are dependent on topography of the understudied hilly area. Topography is the most important feature to be considered in the rockfall and rock-block movement on the critical slopes with debris material [9]. Steep cuts in the rock slope of a transportation corridor are considered most critical for the fall of rock blocks embedded in the soil matrix. If the steep cuts are located in the area of high-intensity rainfall, the rockfall hazard will be at a critical level, as the moisture penetration will trigger the rock-block movement downward to create a hazard [35]. The distance traveled by rock blocks detached from embedded positions can be measured by in situ surveys by identifying marks of the impacts on the toe of the slopes [36].

The permeability of the matrix material in the laboratory is evaluated for the different dry densities and moisture content. The assessment of critical moisture for the specific range of particle shapes can help prevent the failure of the steep cut face of the rock slope. Different moisture content levels result in a wide range of stress–strain curves of the matrix material collected from the field. The evaluation of critical $(MC)_C$ is the most desired parameter for the phenomenon of the triggering of rock-block movement. The shape and gradation of the matrix particles affect the water flow in the soil grains and the shear strength of soil strata [37,38]. The flowing material comprises a mixture of water and sediments of different sizes and shapes [39]. The shape of the particles of debris are mostly extremely irregular, varying from clay to boulder size [40]. The grain size of the matrix material is the crucial aspect for the movement of the rock block embedded in the soil matrix.

The failure of the matrix material in its undrained condition triggers the movement of rock blocks [36]. Matrix flow susceptibility can be assessed effectively by the use of geotechnical approaches, whereas the higher water-retention capacity of soil causes a reduction in the shear strength of the matrix material [36]. Landslides mobilize due to the increase in pore water pressure caused by rainfall in the debris flow susceptibility areas [41].

The elastic modulus $(E_S)$ of soil at the soil–rock interface, i.e., the interface of the rock block and soil matrix, plays a vital role in the triggering of rock-block movement. The $E_S$ of the soil controls the initiation of the movement of the rock block. The $E_S$ parameter depicts the elastic or recoverable strain, whereas the peak shear stress $(P_S)$ presents the failure strength. The triggering of the rock block is strongly dependent on the $E_S$, which shows the stress mobilized before the triggering phenomena.

A peak shear strength test on fine material can be effectively performed in laboratory by a direct shear test. Stress–strain behaviors of matrix soil for $P_S$ are necessary for its characterization to assess the resistive forces to the movement of rock blocks and the soil matrix [42,43]. The $P_S$ of the matrix is the critical factor for the initiation of boulder movement, and causes the ultimate hazard for vehicles at the transport corridor. Rock-block falling is a common phenomenon occurring in cut slopes of mountainous areas where rainfall plays a vital role in the triggering of rockfall, depending upon the amount of rainfall [5]. The inundation of matrix material in lab is performed to simulate the field flooding phenomena for the evaluation of the strength of matrix soils [43]. The soil was compacted at the level of in situ/field dry density and flooded with free water. The moisture governs the residual strength of soil having low cohesion [29]. An important study for interface friction and the loss of shear strength upon wetting of the interface plane has been

conducted [44] on the soil–cement mortar interface, whereas sudden failure can occur at the interface due to the sudden loss of friction [45].

The flow of matrix material with water comprises three phases, i.e., triggering, movement, and accumulation of the soil matrix with rock blocks and fragments, whereas all these phases are difficult to observe in the field [11]. Real rockfall is unlikely to be witnessed; hence, the field studies elaborating the moisture-induced forced rockfall can be carried out to simulate real rockfall events. This simulation phenomenon is very costly and risky in the context of expected hazards in the simulation process. Real-scale rockfall experimentation has been conducted by many researchers in past [46–48].

Past studies have addressed rainfall-induced rockfall; however, the hydrogeotechnical, i.e., moisture-strength factor for rockfall hazards has rarely been studied [49,50]. Previous studies on KKH were conducted [51–53] for factors such as aspect, faults, elevation, curvature, lithology, slope, topography, rainfall intensity, drainage, and geology. There is an absolute need to formulate preventive measures in most critical rockfall hazard areas, i.e., the Tatta Pani section of KKH, to correlate the moisture content to the elastic modulus ($E_S$) and peak shear stress ($P_S$) of matrix soil to predict the critical moisture content $(MC)_c$ causing the triggering of rockfall. In this study, an innovative, economical, and significant predictive approach involving the hydrogeotechnical factor is presented for real-scale moisture-induced forced rockfalls to manage the resources for preventive measures for the reduction in fatalities. Few studies have been seen for the novel hydrogeotechnical initiator factor of matrix softening, i.e., a substantial reduction in Es and Ps of low-cohesive-matrix soil.

The study intends to meet the following objectives:

1. Collection of rock-block samples and soil matrix from a rockfall hazard area of the Tatta Pani site, KKH, and the determination of field dry density $(\gamma_d)_F$ by a large-diameter core cutter inserted in the matrix of a steep cut slope face.
2. Laboratory evaluation of the elastic modulus ($E_S$) and peak shear stress ($P_S$) of the soil matrix at the in situ density condition and variable moisture contents ranging from dry to wet phases, and correlation of the moisture content of the soil matrix to the Es and Ps in laboratory for a specific range of grain sizes and constant roughness of rock, i.e., 0.5 mm for the performance of quick direct shear tests.
3. Validation of critical values of laboratory-based moisture contents, i.e., $(MC)_{LC}$ by performing moisture-induced real-scale forced rockfall triggering in the field by the inundation of matrix soil and determining the field critical moisture content $(MC)_{FC}$.
4. Statistical analysis of the dataset.

This study was focused on the Karakoram Highway (KKH), which is a part of the China–Pakistan Economic Corridor (CPEC). The main studied site was Tatta Pani, which is among the sections highly prone to rockfall hazards. Figure 1 shows embedded rock blocks in matrix soil at the studied site of Tatta Pani. The rockfall hazard value for the study section is around 757 when rated using Pierson's Rockfall Hazard Rating System (RHRS). The average annual rainfall for Tatta Pani is 100 mm and the mean annual temperature is 16.2 °C. The average height of the cut slope face ranges from 200 ft to 225 ft from the road level. The slope angle ranges from 85° to 88°. The study section receives the most rainfall in March and July. For the last two years, severe rockfall at the study site has been observed, which has caused huge traffic blockage, even for up to one to two weeks.

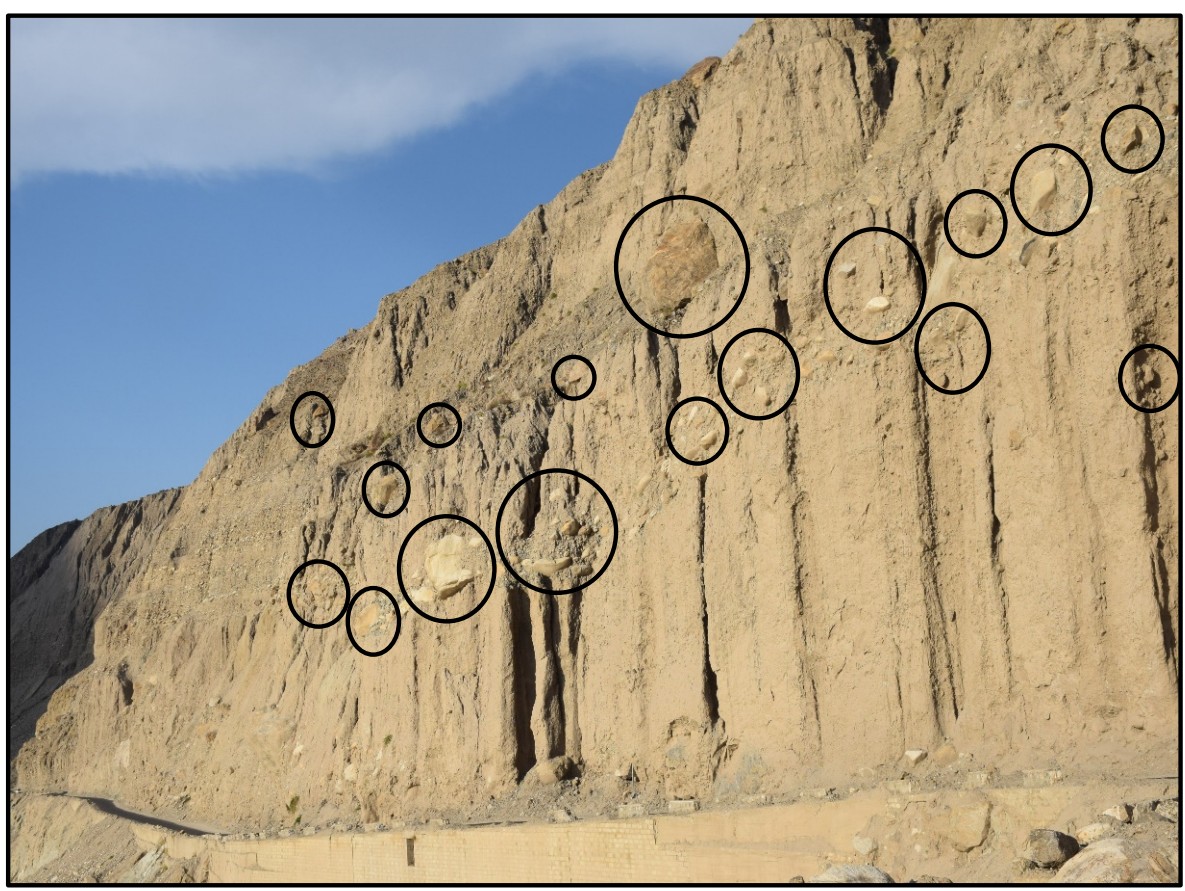

**Figure 1.** A view of scattering pattern of rock blocks in matrix at Tatta Pani, KKH.

## 2. Materials and Methods

Collection of bulk samples of matrix materials and rock blocks was performed at Tatta Pani. The matrix material collected from the steep cut slope along the road comprised mixture of clay, silt, sand, and rock blocks. The size of the matrix material ranging from 6.3 mm to 0.002 mm with embedded rock blocks was studied at the site. Disturbed matrix soil was packed in plastic bags as bulk samples and different rock blocks were collected for the cutting of rock pieces to be used in quick direct shear tests. Samples of matrix soil for the determination of field dry density and field moisture were collected separately by use of large-diameter specially fabricated core cutter.

Field and laboratory studies were performed for evaluation of matrix material. Methodology of this research comprised a field study, including (i) collection of matrix material from study area and (ii) real-scale moisture-induced rockfall at study area; and a laboratory study, comprising (i) classification of matrix material by grain size and (ii) quick direct shear tests at dry and wet conditions for (a) soil–soil and (b) soil–rock interface.

Sampling of matrix and rock blocks from the steep cut slopes is extremely difficult in mountainous areas due to danger of triggering of rock blocks by disturbance created by the sampling method adopted in field. Undisturbed samples of matrix were obtained from study area, and density was determined for use in laboratory testing. Field density tests were performed on the face of the cut slopes by specially manufactured sharp-edged cutting steel molds of 3.5" and 5" diameter with 4" height each, as shown in Figure 2, with area ratio of 8.3%. Two holes of 3 mm diameter were made in the bottom of each mold wall showing the air exit path when air is compressed in the bottom of mold during hammering and insertion of mold into matrix soil. This approach is in line with the core-cutter method outlined in ASTM D2937.

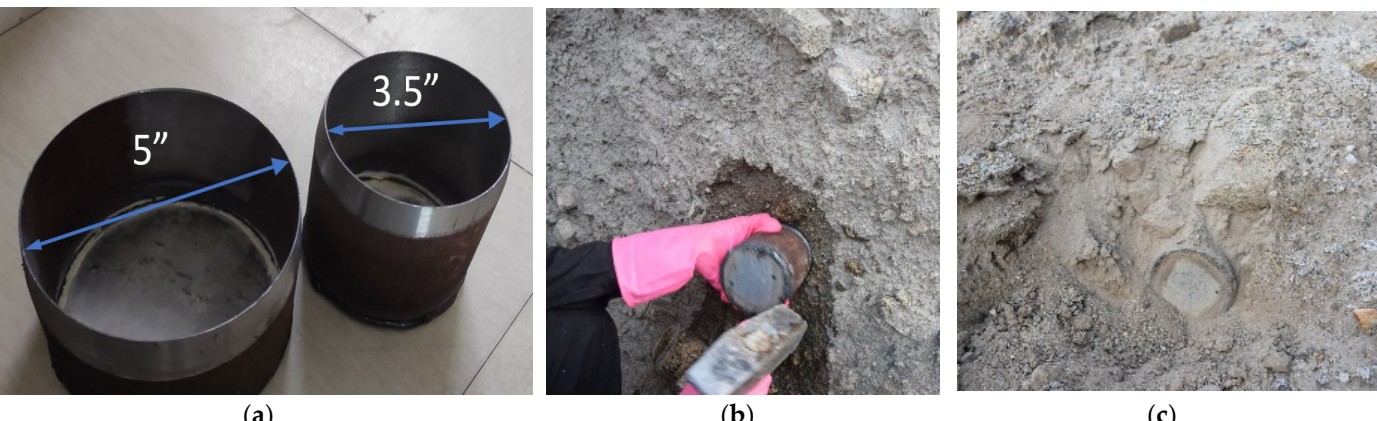

**Figure 2.** (**a**) Large-diameter samplers (**b**) 3.5″ dia. sampling (**c**) 5″ dia. sampling.

The steep face of the cut slope was prepared to insert the mold in horizontal direction. Insertion of molds into the matrix required hammering to fully fill the mold with matrix soil. This technique was observed to be an effective approach for field density determination on critical steep cuts. The field density was simulated in the laboratory for the assessment of peak shear stress (Ps) and elastic modulus ($E_S$) of the matrix material.

The study of moisture-induced rock-block triggering was performed on the blocks with volumes ranging from 0.26 ft$^3$ to 0.31 ft$^3$ and weights ranging from 15.7 kg to 20.3 kg, with embedment volumes of 8% to 12.3%. The embedment of rock blocks was measured by marking the boundary of matrix on the rock blocks at the embedded position of rock blocks in field.

Methods of geotechnical evaluation of matrix material included grain size analysis, Atterberg limit test, moisture content test, the direct shear (DS) test, and unconsolidated undrained (UU) triaxial compression tests for evaluation of peak shear strength at different moisture contents in unsaturated and saturated phases.

Three grain size analysis tests of the representative matrix material show grain size range from 6.3 mm to 0.002 mm. Three Atterberg limit tests on the matrix material were conducted as per ASTM D4318. Consistency parameters such as liquid limit (LL) and plasticity index (PI) were observed in the range of 27% to 28% and 7% to 9%, respectively, for the material passing #40 sieve. The classification of the material as per ASTM D2487 revealed that most of the material falls in the category of low plastic clayey sand.

Moisture content of the disturbed samples was determined by the method outlined in ASTM D2216. The material collected in the plastic jars from the field was tested for moisture content to evaluate the field moisture content of the samples recovered from the steep cut slope before and after rock-block movement. Moreover, moisture content tests were also performed for the remolded samples of matrix material for direct shear tests at different moisture content levels in saturated and unsaturated phases, i.e., wetting–drying conditions.

The quantification of resistive forces is performed by conducting direct shear tests on soil–soil and soil–rock interface. The resistive forces between the soil and rock surfaces are substantially decreased by the quick wetting process. The wetting of cohesive-matrix soil causes the detachment of rock blocks from the embedded position in matrix.

Quick direct shear tests in direct shear apparatus were performed on soil–soil interface and soil–rock interface. The samples were tested on different moisture content levels, including unsaturated and saturated phases of the matrix. The strain rate of 0.2 mm/min was selected for the shearing phase of the direct shear (DS) tests to simulate the sudden undrained shearing of the matrix in rainy season. The peak shear stress ($P_S$) and elastic modulus ($E_S$) of the matrix material is evaluated from stress–strain curve of shearing phase of DS tests. The granite pieces were cut in the size of (6 cm × 6 cm) with 0.5 mm roughness from the rock blocks collected from site to study the soil–rock interface resistive strength, as shown in Figure 3.

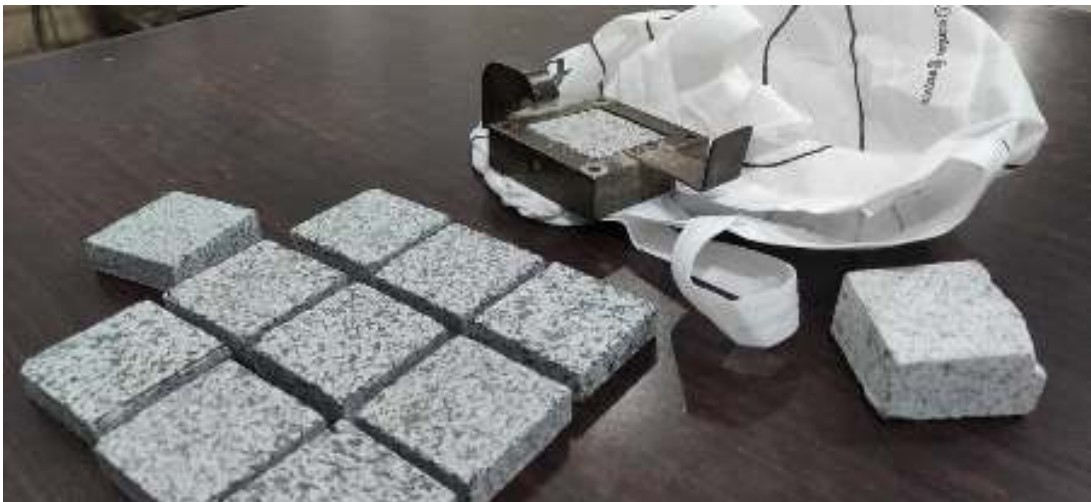

**Figure 3.** The cut pieces of rock blocks for direct shear tests of soil–rock interface resistance.

The seepage at the soil–rock contact surface is compared with the seepage through the soil particles of the matrix. For this purpose, direct shear tests were conducted for the predetermined slippage plane, i.e., the contact surface of soil–rock. The undulations were taken as rough surface of the block in field. The upper portion of the direct shear samples was taken as soil matrix. The soil matrix was taken as the representative of the thirty-two samples collected from the cut slope face selected in field for mixing of different moisture contents, i.e., 1% (taken as dry phase, simulating the field dry samples), 9%, 15%, and 21% (taken as wet phase). Four different moisture contents, i.e., 1%, 9%, 15%, and 21%, were selected in study for evaluation of $P_S$ and Es for a wide range of moistures, starting from dry phase to wet phase of field material. Initially, the remolded samples of field materials, i.e., clayey sand, were prepared, and degrees of saturation (S) were measured for each moisture content level. The moisture content level of 21% was achieved for 100% saturation level for this soil, and this moisture content was considered as the "wet phase moisture" for the experimental soil in this study. It was observed that moisture content larger than 21% revealed difficulty of sample preparation due to start of soil flow. Forty direct shear tests (twenty for soil–soil interface and twenty for soil–rock interface) were conducted, i.e., five tests each for 1%, 9%, 15%, and 21% moisture levels.

Substantial reduction in strength of soil at the soil–rock interface causes the detachment of rock blocks from the imbedded position. The soil–rock interface friction was studied in laboratory, as shown in Figure 4, by quick direct shear tests to simulate the quick undrained failure phenomena at soil–rock interface. The moisture content data show the critical moisture level for the substantial collapse of $P_S$ and Es to be used for assessment of initiation of rockfall triggering in field.

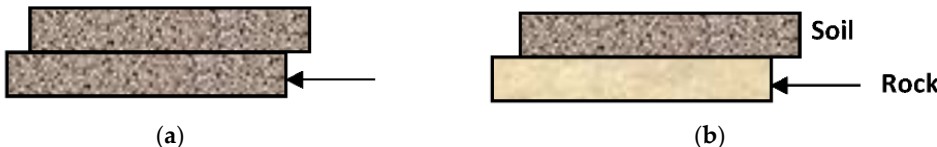

| (**a**) | (**b**) |
|---|---|

**Figure 4.** Direct shear test for (**a**) soil–soil interface resistance and (**b**) soil–rock interface resistance.

The permeability tests on the matrix material were performed to assess the infiltration of water in soil matrix collected from the steep cut slopes of Tatta Pani. Three permeability tests were conducted on the representative matrix soil as per ASTM D2434. The coefficient of permeability (k) of the matrix material was compared with literature to assess the type of soil regarding the range of permeability.

## 3. Results

Presented in Figure 5 is the average grain size analysis curve for the matrix material for the study area. The curve shows that the grain size of the Tatta Pani site ranges from 6 mm to 0.075 mm and 87 % sand, 5% silt, and 8% clay. The gradation curve in Figure 5 shows a large portion of sand (coarser portion of filed soil), which depicts high porosity and permeability. The sand fraction of the field soil governs the flow of water at the soil–rock interface [32]. The porosity of the soil was measured as per the method described by [32] to access the absorption of rainfall during the experimentation of real-scale rockfall. The plasticity index (PI) ranges from 5 to 7, depicting the classification of matrix material as per ASTM D2487 to be SC (clayey sand).

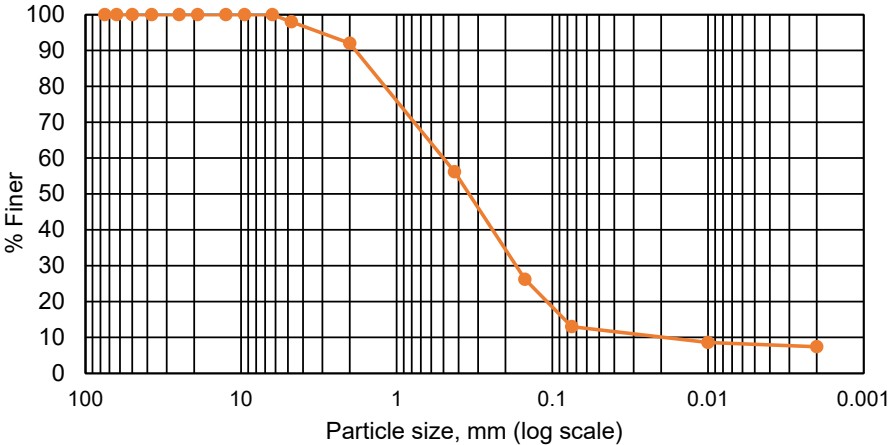

**Figure 5.** Grain size curve of matrix material collected from Tatta Pani.

Typical stress–strain curves are presented in Figure 6 for the soil matrix. The peak shear stress ($P_S$) and elastic modulus ($E_S$) were evaluated by performing 40 direct shear tests at different moisture content levels of matrix soil. The $P_S$ and $E_S$ values were observed to be in the range of 88.83 kPa to 29.5 kPa and 11.83 MPa to 2.1 MPa, respectively, from the soil–soil interface resistance tests for the variation of moisture content from 1% to 21%, whereas the $P_S$ and $E_S$ were observed to be in the range of 67.2 kPa to 7.2 kPa and 8.7 MPa to 1.0 MPa, respectively, in the case of soil–rock interface resistance tests, respectively, at the moisture content range of 1% to 21%. The collapse of $P_S$ and $E_S$ was observed at 21% moisture content, i.e., laboratory-based critical moisture content $(MC)_{LC}$. The correlations of the $P_S$ and $E_S$ are high in the dry phase due to the bonding of the clay fraction (cohesive material) with the sand fraction (non-cohesive material). Clay shows high bonding strength in its dry condition and depicts collapsing behavior in its wet condition [54].

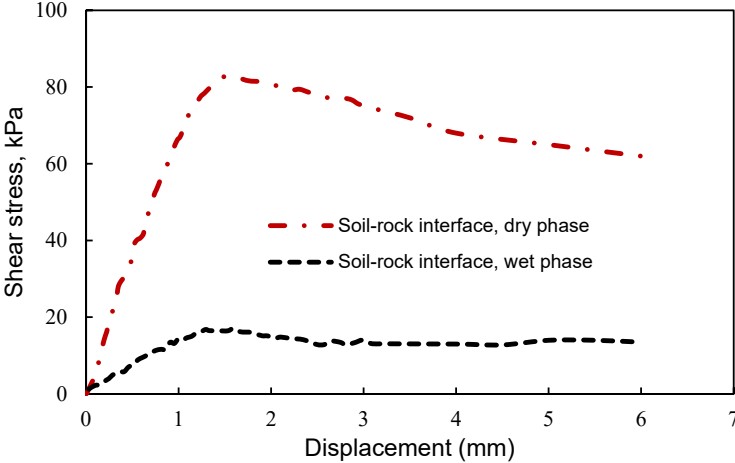

**Figure 6.** Typical stress–strain curves of direct shear test in dry and wet phases of soil–rock interface.

The following correlations of moisture content (MC) with the peak shear stress ($P_S$) (Equations (1) and (2)) and elastic modulus ($E_S$) (Equations (3) and (4)) are developed (Figures 7 and 8) at the matrix density ranging from 1.78 g/cm$^3$ to 1.92 g/cm$^3$ (equivalent to field density) for the soil–soil (Equations (1) and (3)) and soil–rock interfaces (Equations (2) and (4)):

$$P_S = -0.1841(MC)^2 + 1.4277(MC) + 84.009 \tag{1}$$

$$P_S = -0.1838(MC)^2 + 1.4443(MC) + 60.851 \tag{2}$$

$$Es = -0.0221(MC)^2 + 0.0652(MC) + 11.061 \tag{3}$$

$$Es = -0.0165(MC)^2 + 0.002(MC) + 8.268 \tag{4}$$

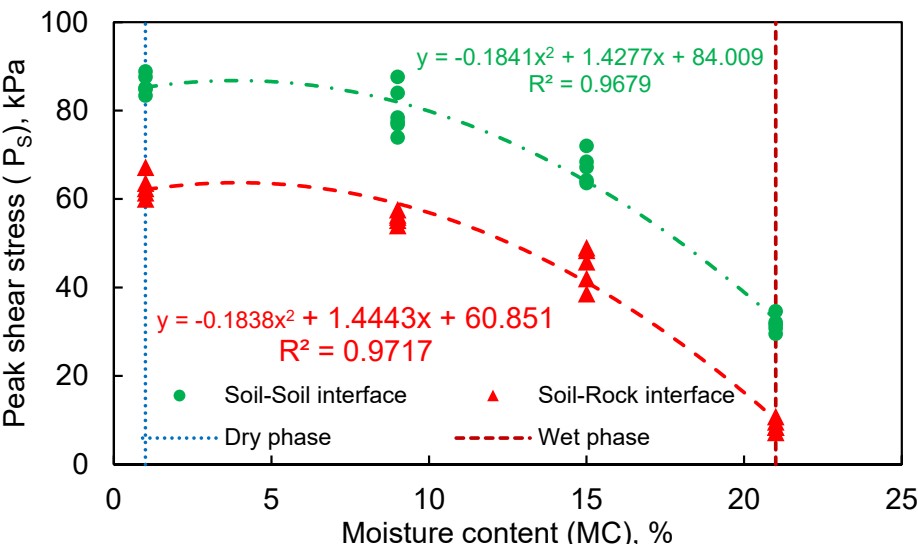

**Figure 7.** $P_S$ of soil−soil and soil−rock interface of Tatta Pani site.

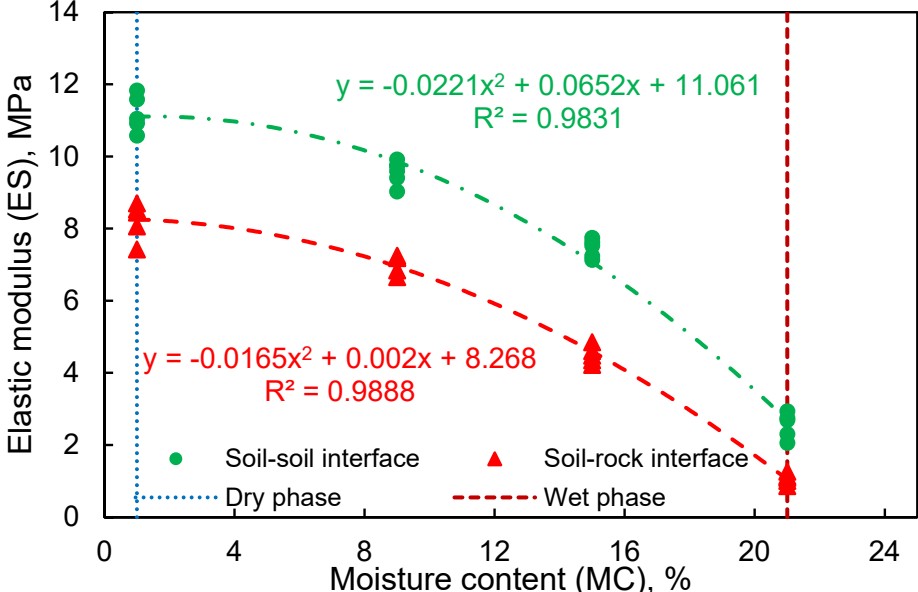

**Figure 8.** $E_S$ of soil−soil and soil−rock interface at Tatta Pani site.

The developed equations (correlations) in the present study have been compared with the literature, and a resemblance of the drop in the shear stress and elastic modulus was observed when compared with the study of [54], which presents a substantial drop in

the shear stress and elastic modulus from 15% to 23% moisture content, validating the data of present study; however, the difference in values is attributed to the difference in material properties.

The substantial collapse of the $P_S$ and $E_S$ causes the triggering of rockfall. The critical values of the $P_S$ and $E_S$ can predict the critical rainfall intensity in the area expected to cause rockfall. Laboratory data on samples collected from the field from forty quick direct shear tests depicted a 3-fold drop in the $P_S$ at the soil–soil interface and a 9.3-fold drop at the soil–rock interface by varying the moisture content from 1% (taken as dry phase) to a critical laboratory moisture content $(MC)_{LC}$ of 21% (taken as wet phase). Similarly, a drop in the $E_S$ was observed to be 5.7-fold at the soil–soil interface and 10-fold at the soil–rock interface, causing a triggering in rockfall in the clayey sand with blocks, i.e., block-in-matrix soil (BIM soil). Hence, the $P_S$ and $E_S$ values for the triggering of rockfall have been mentioned in the text for the soil–rock interface, in addition to the single factor, i.e., critical moisture content.

The quick direct shear tests and unconsolidated undrained triaxial tests were performed in laboratory, due to the fact that the rockfall is a rapid process. The collapse occurrence time was observed in a field simulation test, which helped in the setting of the total time of failure in the direct shear tests, i.e., the strain rate assessment for the direct shear tests.

Physical studies of rockfall are significant for real data analysis and the precise prediction of hazards [55]. A novel approach was applied to saturate the surrounding material of the large blocks for intended fall of boulders. The present study is intended to simulate rockfall at Tatta Pani and the laboratory studies of the moisture strength relation of the matrix material of Tatta Pani. To devise the field experimentation of moisture-induced rockfall, the goals of the field study were set. As the porosity of the soil is a very important factor in study of moisture-induced rockfall, the porosity of the field soil material was determined in the laboratory having an average value of 0.31. The water was showered on the periphery of the embedded rock blocks to make the matrix soil in a fully saturated condition. Rock blocks of about 12–16 inches were selected for forced fall by showering water from upside of the rock blocks. It was observed that the water penetrated the soil–rock interface for a duration of 2 to 3 h and forced the triggering of rock-block movement.

Figures 9 and 10 show the phenomena of rockfall from the embedded position in the soil matrix, i.e., the case of block-in-matrix soils. In the field, it can be observed that the rock blocks embedded in the soil matrix are retained in place due to the peak shear strength and non-exceeding critical moisture $(MC)_C$ of the matrix material. The rockfall is mainly caused due to the reduction in the shear strength of the matrix material by the increase in moisture content; in particular, when the shear strength of the soil mass drops to a level lower than the sliding force during slope failure, it ultimately results in a disaster with fast soil flow, i.e., debris flow in hilly areas [3,56,57]. Due to rainfall in the mountainous area, the water infiltrates into the matrix material and pore water pressure increases, resulting in a 90% chance of loss of shear strength in the soil–rock mixture (SRM), causing landslide occurrence [13,58].

The failure pattern of the matrix in the field depicts the movement of different grain size particles of the matrix in the field. Water flow patterns in the area depict the availability of the critical quantity of water required for the initiation of boulder movement. The water flow channels and the rainfall intensity of the area affect the erosion of the matrix material of the steep cut slopes in the study areas. The rock-block movement pattern observed in the field shows that the fine material at the surface of the rock-block loses its shear strength and collapses, resulting in the initiation of rock-block movement, and a rockfall hazard takes place in the field.

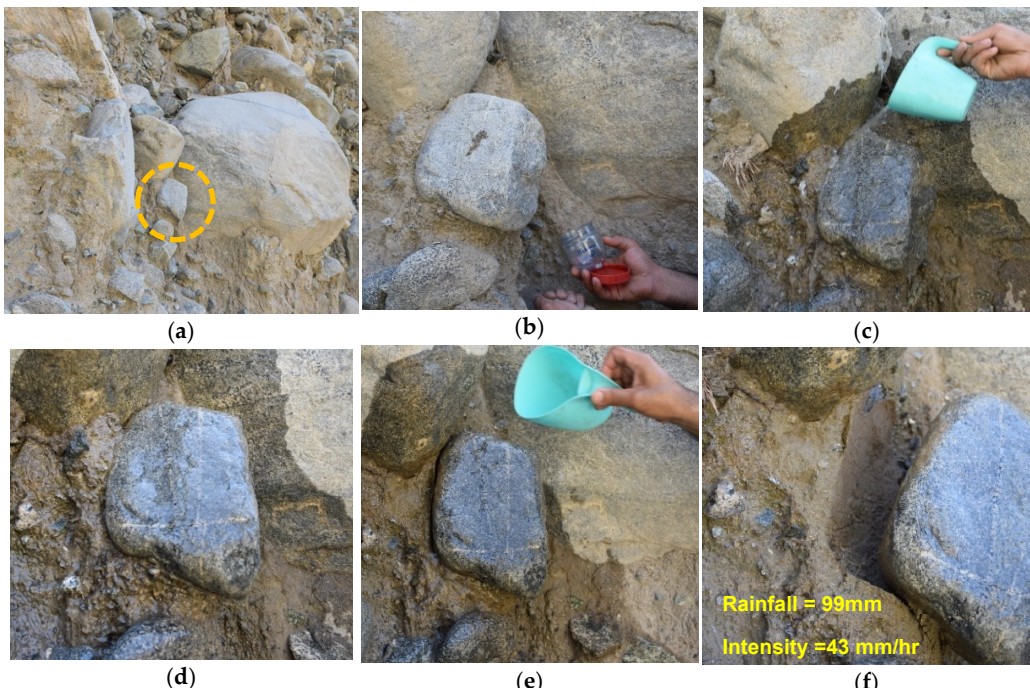

**Figure 9.** Stages of real-scale rockfall at Tatta Pani section KKH (Location 1) (**a**) Rock-block (**b**) Matrix collection (**c**) inundation started (**d**) initiation of detachment of rock-block (**e**) widening of soil/rock-block gap (**f**) occurrence of rockfall.

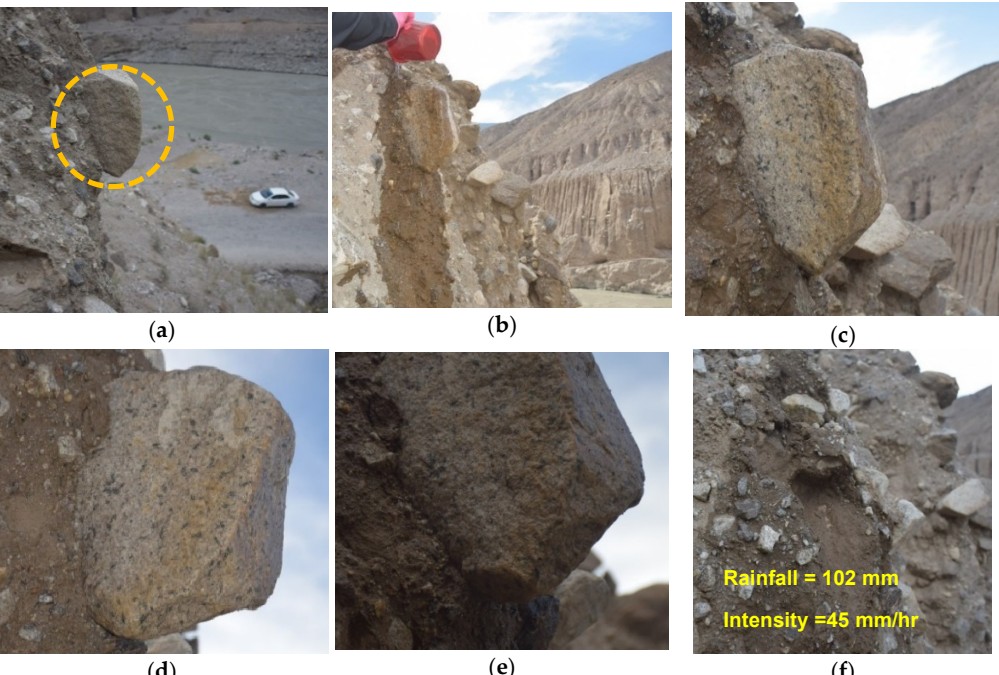

**Figure 10.** Stages of real-scale rockfall at Tatta Pani section KKH (Location 2) (**a**) Rock-block (**b**) inundation started (**c**) Initiation of rock-block detachment (**d**) Enlargement of soil/ rock-block gap (**e**) Widened soil/rock-block gap (**f**) Occurrence of rockfall.

Thirty-two real-scale moisture-induced rockfalls were performed in the field (after stopping traffic on both sides). The simulated watering in the field was applied near the upper line of the soil–rock contact to assure the total water absorption at the soil–rock contact surface. The rate of water falling was related to the rainfall intensity for rockfall expected in the area. The critical moisture contents that caused rockfall in Figures 9 and 10

were observed to be 21.5% and 22.7%, respectively, whereas the corresponding critical rainfall intensities were found to be 99 mm and 102 mm, respectively. Soil moisture sensors were used to monitor the field moisture content during the moisture-induced forced rockfall at the site. Furthermore, the samples of soil in the dry and wet phase (at the rockfall stage) were checked for moisture by use of a speedy moisture tester for validation in the field. In addition, the soil samples taken from the soil contact surface after rockfall were packed in airtight plastic containers for moisture content tests in laboratory. The critical moisture content range of the soil matrix collected from the field, i.e., $(MC)_{FC}$ at the rockfall phase, were observed to be 20% to 23%, which is very near to the critical moisture content $(MC)_{LC}$ determined in laboratory, i.e., $(MC)_{FC}$ is within 20 % of $(MC)_{LC}$. Hence, it is inferred that direct shear tests conducted in laboratory for the soil–rock interface resistance can be effectively used for the prediction of the collapse of shear strength and the elastic modulus in the case of rockfall for block in-matrix soils. It was observed in the field that the wetting of the contact surface caused the triggering of rockfall instead of the additional wetting of the surrounding area of the rock block. Hence, the moisture content of the soil in field was determined for the central portion of the soil patch, i.e., the slippage surface for the rock block, as shown in Figures 9f and 10f.

Analysis of variance (ANOVA) of datasets of the field and laboratory results was conducted, and *p*-values showed that data were found in good health, as no outlier was found in the data, including laboratory and field experimentation.

The field research in rockfall areas is always risky and costly. This study covered only the extremely critical steep cut slopes to study the fall of boulders or rock blocks. Almost the same sizes of blocks ranging from 0.26 ft$^3$ to 0.31 ft$^3$ and weights ranging from 15.7 kg to 20.3 kg were selected to analyze the variation of $P_S$ and $E_S$ with the increase in moisture content of the soil–rock interface in the field. The same phases of unsaturated and saturated matrix were used in laboratory. However, the large-volume blocks were not covered in this study due to expected severe hazard creation on the adjacent transportation corridor; however, these aspects should be covered in future research. It is also noteworthy that the steep slope at the Tatta Pani site with a different matrix soil mixture should also be studied in future to cover a full range of matrix materials for a comprehensive rockfall hazard protection model.

Although the slope angle, weight of rock block, size of rock block, and matrix density at the soil–rock interface play a role in the quantification of rockfall, the critical moisture content is observed to be a significant factor in the triggering of rockfall. In this study, the real-scale moisture-induced forced rockfall was triggered by inducing moisture at the soil–rock interface by selecting a very narrow range of slope angles, the weight of the rock block, the size of the rock block, and the matrix density at the soil–rock interface to critically observe the effect of single parameter, i.e., the critical moisture content $(MC)_C$ of the matrix at the soil–rock interface. Although the developed correlations are parameter-specific and site-specific, the proposed innovative methodology of field inundation in comparison with laboratory data can be used to develop predictive correlations to prevent/reduce fatalities near vertical rock cut slopes in KKH.

A thirty-liter-capacity container was arranged for the showering test. Showering was conducted by a small pot of known volume. The water showering was applied on a 2 ft × 2 ft (4 sft) area, including the face of the rock block. Showering was continued until the fall of the rock block. The net volume of water was noted as the critical moisture causing rockfall. The total volume of the water used in showering for each of the 32 rockfall-triggering cases was converted into millimeters of rainfall, having a range from 97 mm–107 mm of water (simulated rainfall amount is shown in Figures 9 and 10), which falls in the annual precipitation of the Tatta Pani site, KKH, i.e., 112 mm. Keeping in view the historical event-based precipitation in the study area, the simulated rainfall can be considered too high. This hydrogeotechnical study indicates that the critical moisture $(MC)_C$ (converted in mm of rainfall) can well predict the anticipated triggering of rockfall,

which can be assessed by the rainfall prediction of the site so that the time/day for stoppage of traffic may be decided to prevent fatalities.

## 4. Conclusions

A hydrogeotechnical study for the prediction of rockfall triggering was conducted at a critical cut slope face of the rockfall hazard area of the Tatta Pani site, KKH, with cut slope angles ranging from 85° to 88°, rock-block volumes ranging from 0.26 ft$^3$ to 0.31 ft$^3$, and weights ranging from 15.7 kg to 20.3 kg for moisture-induced forced rockfall. The following conclusions can be drawn from this research:

1.  Quick direct shear laboratory data showed a 3-fold collapse in peak shear stress (P$_S$) at the soil–soil interface and a 9.3-fold drop at the soil–rock interface by varying the moisture content from 1% to a critical laboratory moisture content (MC)$_{LC}$ of 21%. Similarly, there was a 5.7-fold drop in the elastic modulus (E$_S$) at the soil–soil interface and a 10-fold decrease at the soil–rock interface for the variation of moisture content from 1 % to 21% for the low-cohesive-soil matrix with a permeability range of $3 \times 10^{-4}$ to $5.6 \times 10^{-4}$ m/s. Hence, the (MC)$_{LC}$ of 21% depicted the substantial loss of P$_S$ and Es at the soil–rock interface.
2.  Hydrogeotechnical relations, i.e., moisture content (MC) versus the peak shear stress (P$_S$) and elastic modulus (E$_S$), at the soil–rock interface are developed for the prediction of rock by using MC as the critical factor, which causes the collapse of P$_S$ and E$_S$ in the saturated phase of soil at the soil–rock interface.
3.  A proposed laboratory approach for the collapse of P$_S$ and E$_S$ is validated by an innovative field inundation of the soil–rock interface to cause moisture-induced real-scale rockfall triggering at field dry density ($\gamma_d$)$_f$ and a critical field moisture content (MC)$_{FC}$ of the matrix ranging from 1.78 g/cm$^3$ to 1.92 g/cm$^3$ and 1.3% to 25.4%.
4.  Variation in (MC)$_{FC}$ was within 20% of (MC)$_{LC}$, which depicts a good confidence level for using proposed correlations in matching the lithology of block-in-matrix soil. It was also observed that the E$_S$ and P$_S$ values of the soil–rock interface are substantially lower as compared with the soil–soil interface.
5.  The outcome of hydrogeotechnical studies for 32 rockfall cases also revealed that the triggering of rockfall may start at rainfall ranges from 97–107 mm, falling in the intense rainfalls of the monsoon and torrential period of the Tatta Pani site, KKH.

**Author Contributions:** Conceptualization, E.M.; methodology, I.R. and A.T.; formal analysis, E.M.; investigation, E.M.; resources, A.M.E.; data curation, K.F.; writing—original draft, E.M.; writing—review and editing, A.M.E.; visualization, F.A.; supervision, I.R. and K.F.; project administration, I.R.; funding acquisition, A.M.E. All authors have read and agreed to the published version of the manuscript.

**Funding:** This research received no external funding.

**Conflicts of Interest:** The authors declare no conflict of interest.

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
