# Peer review of "Hydrogeotechnical Predictive Approach for Rockfall Mountain Hazard Using Elastic Modulus and Peak Shear Stress at Soil–Rock Interface in Dry and Wet Phases at KKH Pakistan"

_sustainability, doi:10.3390/su142416740_

Round 1

Reviewer 1 Report

1.      Line 275: Why did you choose the four different soil moisture for testing, 1% (taken as dry phase, simulating the field dry samples), 9 %, 15% and 21 % (taken as wet phase)? How did you identify the saturated soil and unsaturated soil?

2.      Line 318: The correlations of moisture content (MC) with peak shear stress (PS) (eq. 1 and eq.2) and elastic modulus (ES) (eq.3 and eq.4) were very high in Fig. 7 and Fig. 8. However, the correlations of wet phase were a litter bit better than that of dry phase. Why?

3.      Line 339: This was a field experimentation of moisture-induced rockfall. The flow of moisture in soil is very important. Could you tell me the average soil porosity of coarse pore?

4.      Line 345: Figs.9 and 10 shows the phenomena of rockfall from the embedded position in soil matrix i.e., the case of block-in-matrix soils. Please show the simulated rainfall amount and rainfall intensity in Figs. 9 and 10. In addition, please show the annual precipitation in the study area. Are the simulated rainfall amounts too high in both Figs. 9 and 10?

5.      Line 363: What is the critical point of soil moisture or rainfall intensity for rock falling in Figs. 9 and 10?

6.      Line 368: The critical moisture content range of soil matrix collected from field i.e., (MC)FC at the rockfall phase were observed to be 20% to23%. How to monitoring the soil moisture content in field? Did you establish the soil moisture sensors?

7.      This hydro-geotechnical study for prediction of rockfall triggering was useful and helpful for monitoring rockfall hazard area, especially for semi-arid area.

Reviewer 2 Report

The research has certain engineering value.

Also need to supplement the references, research content introduction.

The authors are asked to add the shortcomings of the study in the conclusion.

Relevant formulas need to be compared with relevant references.

Round 2

Reviewer 2 Report

The author carefully revised all the questions.

The author responded to all comments.

The quality of the manuscript has been greatly improved.

It is recommended to accept and publish.
